# Association between over-indebtedness and antidepressant use: A cross-sectional analysis

**Jacqueline Warth**[1]*, **Niklas Beckmann**[1], **Marie-Therese Puth**[1,2], **Judith Tillmann**[1], **Johannes Porz**[1], **Ulrike Zier**[1], **Klaus Weckbecker**[1,3], **Birgitta Weltermann**[1], **Eva Münster**[1]

**1** Institute of General Practice and Family Medicine, University of Bonn, Bonn, Germany, **2** Department of Medical Biometry, Informatics and Epidemiology (IMBIE), University Hospital Bonn, Bonn, Germany, **3** Faculty of Health/ Department of Medicine, Institute for General Medicine and Interprofessional Care, University Witten/Herdecke, Herdecke, Germany

* jacqueline.warth@ukbonn.de

## Abstract

### Background

Burden of disease caused by depression and its association with socioeconomic status is well documented. However, research on over-indebtedness is scarce although millions of European citizens in all socioeconomic positions are over-indebted. Prior studies suggested that over-indebtedness is associated with poor physical and mental health.

### Aims

Investigate the association between over-indebtedness and antidepressant use in Germany.

### Method

A cross-sectional survey among debt advice agencies' clients was conducted in North Rhine-Westphalia, Germany, in 2017 (OID). Data were merged with the first wave of the German Health Interview and Examination Survey for Adults (DEGS1). Descriptive statistics and logistic regression analysis were used to examine antidepressant use in the previous 7 days (OID: n = 699; DEGS1: n = 7115).

### Results

Prevalence of antidepressant use was higher in the over-indebted (12.3%) than the general population (5.0%). The over-indebted were significantly more likely to use antidepressants than the general population even after controlling for other socioeconomic, demographic and health factors (adjusted odds ratio 1.83; 95% confidence interval 1.35–2.48).

### Conclusions

Stakeholders in health care, debt counselling, research and social policy should consider the link between over-indebtedness and mental illness to advance the understanding of health inequalities and to help those who have mental health and debt problems.

**Data Availability Statement:** The data on over-indebted participants (OID survey) are not publicly available as these contain potentially identifying participant information that could compromise

participants' privacy. Therefore, data requests may be sent to the ethics committee of the Medical Faculty of the University of Bonn (Email: ethik@uni-bonn.de). Data from the general population survey (DEGS1) conducted by the Robert Koch Institute is available to use by the scientific community on application as Public Use Files. Data can be requested upon application from the Research Data Centre: https://www.rki.de/EN/Content/Health_Monitoring/Public_Use_Files/public_use_file_node.html.

**Funding:** Project manager (EM), OID survey - Landeszentrum Gesundheit NRW - Centre for Health North Rhine-Westphalia, Germany - https://www.lzg.nrw.de/. The funders had no role in study design, data collection and analysis, decision to publish, or preparation of the manuscript.

**Competing interests:** The authors have declared that no competing interests exist.

## Introduction

Recent estimates suggest that about 10 percent of households across EU countries are over-indebted and numbers continue to increase [1,2]. In Germany, 6.9 million individuals are considered over-indebted, thus unable to cover payment obligations and living expenses with available assets on an ongoing basis [3]. Over-indebtedness may reflect both a cause and consequence of poor health [4].

Research on social determinants of health has a long tradition but most studies have focused on socioeconomic status (SES) measures including education, income and occupation [5]. However, individuals across the socioeconomic spectrum may face over-indebtedness which cannot be considered interchangeable with the standard socioeconomic measures [2,5]. Several studies have shown that measures of over-indebtedness are associated with various health outcomes, independent of standard SES measures [6]. Besides physical morbidity, e.g. diabetes [7], obesity [8] and pain [9,10], over-indebtedness has consistently been found to be associated with mental illness, including psychoses [7] and depression [11–17].

Prescriptions of antidepressants continue to increase worldwide [18,19]. While depressive disorders are the main indication for antidepressant prescriptions, antidepressants are also prescribed for other indications such as anxiety, sleeping disorders and pain [20,21]. Thus, antidepressant use may not only serve as an indicator of mental illness but also provide a first insight into access to mental health care. Depressive disorders are currently one of the three leading causes of disability globally and the associated disease burden continues to increase [22,23]. A large body of evidence suggests a social gradient in the prevalence of depression: Those in low socioeconomic positions are more frequently affected by depression [24–26]. Some studies have suggested an association between SES measures and antidepressant use but evidence is inconsistent [27–30]. Previous studies mainly examined US data. However, there are variations in legal as well as social consequences of over-indebtedness across countries that may affect the prevalence of mental illness and medication use considerably. Therefore, the aim of the present explorative study was to investigate the association between over-indebtedness and antidepressant use in Germany.

## Materials and methods

To evaluate the association of over-indebtedness and use of antidepressants, we cross-sectionally compared a sample of clients of debt advice agencies (OID survey) to a secondary data sample of the German general population from the first wave of the German Health Interview and Examination Survey (DEGS1).

### Data

We pooled data from the following two sources for the analysis.

In the OID survey ("medication use and self-medication in the over-indebted"; German acronym: ArSemü), clients of debt advice agencies received a standardised questionnaire and a postage-paid return envelope from their counsellor that could be returned to the study centre anonymously. Recruitment took place between July and October 2017 in the German federal state of North Rhine-Westphalia (NRW). Non-profit debt advice agencies were enlisted as recruiters in the study after declaring intent to participate. All debt advice agencies in the state associated with either the local German Consumer Organisation or one of the member organisations of the "Specialist Counselling Debt Counselling NRW" group were invited to act as recruiters by their respective umbrella organizations. The target population consisted of those clients that (a) did not live in the same household as other participants, (b) were at least 16 years of age and (c) had visited the debt advice agency at least twice. The latter criterion was

designed to prevent disruption of initial debt consultations of clients. All questionnaires returned within one month after the end of recruitment were included. Respondents were provided information on study procedures, voluntariness and confidentiality, and confirmed their willingness to participate in the study by returning the questionnaire. The Ethics Committee of the University Medical Faculty in Bonn, Germany, approved the OID-survey (No. 167/17).

The first wave of the German Health Interview and Examination Survey (DEGS1) was designed to serve general-purpose national health monitoring in Germany and was conducted by the Robert Koch Institute, the national public health agency, with the data partly being made available for public use (n = 7987) [31]. In DEGS1, adults from the German general population that had replied to a mailed invitation or were successfully followed up, were interviewed by phone, filled out questionnaires and (mostly) underwent physical examination at local study centres between November 2008 and 2011. The target population consisted of adults aged 18–79 years with permanent residence in Germany. The sampling frame was arrived at by randomly sampling local population registries and the follow-up population of a precursor study (German National Health Interview and Examination Survey 1998, GNHIES98) in two stages subsequently stratifying by municipality and age. The size of the survey was predetermined by power analyses. Participants provided written informed consent. The Charité-Universitätsmedizin Berlin Ethics Committee approved the DEGS1 study protocol (No. EA2/047/08). Further details have been reported elsewhere [32].

We excluded those subjects from our analysis who did not participate in the standardised interview on medication use in DEGS1 (n = 872) as this comprises information on the use of antidepressants.

## Variables

OID participants were asked which, if any, medications they had used within the last 7 days. Medical experts classified medications according to the Anatomical Therapeutic Chemical Classification System (ATC) taking given information on generic or brand or class name and medical indication into account. In DEGS1, participants presented the packages of any medications used in the last 7 days at their physical examination. Medications were then automatically coded according to ATC classes. Missing data was followed up by phone or mail [32]. Based on these data, in both surveys, antidepressant use was assumed for a participant if any of the medications used were classified as ATC class "N06A".

Over-indebtedness was assumed for all participants from the OID survey. Data on confounders was gathered by questionnaire (OID) and phone interview (DEGS1). Age was categorised because the public dataset for DEGS1 did not include a continuous age variable and AIC (Akaike Information Criterion) showed categorisation to be vastly advantageous for the OID survey data. ISCED (International Standard Classification of Education) education level (low, medium, high) of OID participants was derived from questions about the highest level of school qualification and the highest level of tertiary education/vocational training. The latter items developed for the OID survey were similar but less detailed than those used in DEGS1 were. Marital status was recorded using the same item in both surveys. "Previously married" combines divorced, widowed, or separated. Participants of both surveys were asked whether they suffered from any chronic disease with the option of answering "I don't know" (considered missing). A chronic disease was assumed for OID participants if their self-reported medication regimen could be attributed to a chronic disease according to medical experts. Different items were used to measure employment status in the two surveys. Participants who indicated any kind of full or part-time employment including marginal employment in the OID survey

were classified as currently employed. If employment status was not recorded for OID participants, receiving unemployment benefits was assumed to indicate non-employment. In DEGS1, participants whose employment status was summarised as currently employed were compared to the previously or never employed.

## Statistical methods

Statistical analyses were carried out using R (3.5.2) [33]. To assess the association between antidepressant use, over-indebtedness and multiple confounders, crude and adjusted odds ratios (OR) and 95% confidence intervals (CI) were estimated using multiple logistic regression analysis. Multiple imputation was used to handle missing data assuming missingness at random. Based on all and only the variables included in the regression model reported below, 30 datasets were generated and pooled using the "mice" package for R.

## Results

In the OID survey, debt advisors handed out the study material to 1393 eligible clients. A total number of 699 individuals returned the questionnaire with complete data on sex and age, which reflects a response rate of 50.2%. In DEGS1, the response rate was 42% among newly recruited participants and 62% among those who had already participated in GNHIES98 [32].

The merged sample comprised 7814 participants, of which 699 were over-indebted clients of debt advice agencies in the German federal state of North Rhine-Westphalia (OID survey), and 7115 adults living in Germany that participated in the national health survey DEGS1. Females represented 52.3% of the study population (Table 1). Participants of the OID survey were younger and had a lower educational level compared to DEGS1 participants. The majority of over-indebted participants was below the age of 45 (OID survey: 53.3%; DEGS1: 35.1%) and had a low or medium educational level (OID survey: 93.6%; DEGS1: 67.2%). The over-indebted were more frequently unemployed (OID survey: 45.8%; DEGS1: 38.0%), less often married (OID survey: 22.3%; DEGS1: 62.9%) and reported chronic diseases more often (OID survey: 59.9%; DEGS1: 31.8%) than DEGS1 respondents.

In the merged sample, the prevalence of antidepressant use in the last 7 days was 5.6% (Table 2). Prevalence was higher among the over-indebted (12.3%) than the general population sample (5.0%).

Logistic regression analysis found an association between over-indebtedness and use of antidepressants (aOR 1.83; 95% CI 1.35–2.48) that remained significant after adjustment for sociodemographic (i.e. age, sex and marital status), standard socioeconomic measures (educational level and unemployment) and chronic disease (Table 3). Those above the age of 45 were more likely to report antidepressant use than the youngest age group. Females (aOR 2.27; 95% CI 1.81–2.84) had higher odds of antidepressant use than males. Respondents who self-reported chronic disease were nearly four times more likely to use antidepressants (aOR 3.62; 95% CI 2.86–4.57). Individuals who were previously married had greater odds of antidepressant use than the married (aOR 1.39; 95% CI 1.09–1.78). There was evidence of an association between unemployment (aOR 1.67; 95% CI 1.31–2.13) and antidepressant use whereas the association with educational level was not significant after adjustment.

## Discussion

The present study is the first to show increased odds of antidepressant use (aOR 1.83; 95% CI 1.35–2.48) among the over-indebted compared to the general population in Germany. An essential finding was that this association remained significant even after adjustment for

**Table 1. Characteristics of participants.**

| Variable | Total (n = 7814) | | DEGS1[a] (n = 7115) | | OID survey[b] (n = 699) | |
|---|---|---|---|---|---|---|
| Age (years) (n, %) | | | | | | |
| 18–29 | 1187 | 15.2 | 1070 | 15.0 | 117 | 16.7 |
| 30–44 | 1689 | 21.6 | 1433 | 20.1 | 256 | 36.6 |
| 45–64 | 3039 | 38.9 | 2759 | 38.8 | 280 | 40.1 |
| 65–79 | 1899 | 24.3 | 1853 | 26.0 | 46 | 6.6 |
| Sex (n, %) | | | | | | |
| Male | 3726 | 47.7 | 3410 | 47.9 | 316 | 45.2 |
| Female | 4088 | 52.3 | 3705 | 52.1 | 383 | 54.8 |
| Education level (ISCED) (n, %) | | | | | | |
| Low | 1310 | 16.8 | 1006 | 14.1 | 304 | 43.5 |
| Medium | 4131 | 52.9 | 3781 | 53.1 | 350 | 50.1 |
| High | 2314 | 29.6 | 2280 | 32.0 | 34 | 4.9 |
| Missing | 59 | 0.8 | 48 | 0.7 | 11 | 1.6 |
| Chronic disease (n, %) | | | | | | |
| No | 4714 | 60.3 | 4468 | 62.8 | 246 | 35.2 |
| Yes | 2684 | 34.3 | 2265 | 31.8 | 419 | 59.9 |
| Missing | 416 | 5.3 | 382 | 5.4 | 34 | 4.9 |
| Marital status (n, %) | | | | | | |
| Married | 4633 | 59.3 | 4477 | 62.9 | 156 | 22.3 |
| Previously married | 1247 | 16.0 | 978 | 13.7 | 269 | 38.5 |
| Never married | 1845 | 23.6 | 1586 | 22.3 | 259 | 37.1 |
| Missing | 89 | 1.1 | 74 | 1.0 | 15 | 2.1 |
| Employment status (n, %) | | | | | | |
| Employed | 4562 | 58.4 | 4204 | 59.1 | 358 | 51.2 |
| Not employed | 3025 | 38.7 | 2705 | 38.0 | 320 | 45.8 |
| Missing | 227 | 2.9 | 206 | 2.9 | 21 | 3.0 |

[a]General population sample, Germany (2008–2011)
[b]Over-indebted sample, Germany (2017)

standard socioeconomic measures (i.e. educational level and unemployment), sociodemographic and health factors.

Research shows that the findings of the present study may be attributable to higher psychological morbidity in the over-indebted. In a recent national survey among adults in England, the prevalence of common mental disorders (CMD) was 38% among the over-indebted [15]. Our results are consistent with previous studies that demonstrated an association of over-

**Table 2. Prevalence of antidepressant use in the last 7 days.**

| Variable | Total (n = 7814) | | DEGS1[a] (n = 7115) | | OID survey[b] (n = 699) | |
|---|---|---|---|---|---|---|
| Antidepressant use in last 7 days (n, %) | | | | | | |
| No | 7277 | 93.1 | 6736 | 94.7 | 541 | 77.4 |
| Yes | 441 | 5.6 | 355 | 5.0 | 86 | 12.3 |
| Missing | 96 | 1.2 | 24 | 0.3 | 72 | 10.3 |

[a]General population sample, Germany (2008–2011)
[b]Over-indebted sample, Germany (2017)

**Table 3. Association of over-indebtedness and antidepressant use (n = 7814).**

| Variable | Unadjusted OR | 95% CI | Adjusted OR | 95% CI |
|---|---|---|---|---|
| Over-indebtedness | | | | |
| No | Reference | – | Reference | – |
| Yes | 2.86 | 2.22–3.67 | 1.83 | 1.35–2.48 |
| Age (years) | | | | |
| 18–29 | Reference | – | Reference | – |
| 30–44 | 1.76 | 1.08–2.87 | 1.6 | 0.94–2.71 |
| 45–64 | 4.18 | 2.71–6.43 | 3.32 | 2.0–5.5 |
| 65–79 | 3.48 | 2.22–5.44 | 1.84 | 1.06–3.2 |
| Sex | | | | |
| Male | Reference | – | Reference | – |
| Female | 2.44 | 1.97–3.03 | 2.27 | 1.81–2.84 |
| Education level (ISCED) | | | | |
| Low | Reference | – | Reference | – |
| Medium | 0.78 | 0.61–1.0 | 1.01 | 0.77–1.32 |
| High | 0.58 | 0.44–0.78 | 0.95 | 0.69–1.31 |
| Chronic disease | | | | |
| No | Reference | – | Reference | – |
| Yes | 4.91 | 3.94–6.11 | 3.62 | 2.86–4.57 |
| Marital status | | | | |
| Married | Reference | – | Reference | – |
| Previously married | 2.17 | 1.74–2.72 | 1.39 | 1.09–1.78 |
| Never married | 0.71 | 0.54–0.94 | 1.19 | 0.85–1.66 |
| Employment status | | | | |
| Employed | Reference | – | Reference | – |
| Not employed | 2.19 | 1.79–2.67 | 1.67 | 1.31–2.13 |

indebtedness and depression independent of other standard socioeconomic measures such as income, education, and employment status [12,13,16]. Likewise, our findings are in line with those of other recent studies that found an association of over-indebtedness and common mental disorders including anxiety which also reflect common indications for antidepressant prescriptions [15,11,17].

Previous studies have examined the association of diverse measures of over-indebtedness, such as self-reported problems repaying debts [12,11,17] or mortgage arrears [11,13,16] and mental health outcomes. Longitudinal analyses based on the British Household Panel Survey (BHPS) documented that over-indebtedness was associated with worse mental health outcomes including anxiety and the GHQ Caseness Score [11]. In a cross-sectional sample of adults in England, those who were in arrears in utilities, housing or shopping had an increased likelihood of common mental disorders, irrespective of the sources of debt. More specifically, the over-indebted were twice as likely to have a depressive disorder (aOR 2.36, 95% CI 1.59–3.50) and generalised anxiety disorder (aOR 2.51, 95% CI 1.85–3.41) [15]. Likewise, evidence from the South East London Community Health (SELCoH) study showed a significant increase in the odds of CMD among over-indebted respondents over time [17]. Moreover, the latter study found an association between continuous over-indebtedness and talking therapy use in the past year even after adjustment for sociodemographic and socioeconomic variables and prior mental health.

Yet, in the present study the association between over-indebtedness and antidepressant use remained significant even after adjusting for chronic disorders which comprise mental illness.

Hence, increased antidepressant use in the over-indebted might also reflect factors other than psychological morbidity. On the one hand, some studies suggest that prescribing bias might lead to a higher likelihood of pharmacological treatment for patients with lower socioeconomic status [30,34,35]. However, available evidence of an association between standard SES measures and antidepressant use is inconsistent [27–30].

On the other hand, previous studies indicate that individuals with poor mental health are significantly more likely to report cost-related medication nonadherence (CRN) such as not filling prescriptions, skipping doses or splitting pills [36]. Likewise, over-indebtedness has been found to be associated with CRN [37,38,39,13]. In a nationally representative study of Americans older than 50 years, 32.1 percent of those considered over-indebted reported CRN [13]. Among over-indebted individuals that participated in the present study, the prevalence of CRN was 33.6% [40]. The chronically ill were found to have significantly greater odds of CRN than those without a chronic illness. In Germany, co-payments of 5 to 10 euros are required for each prescribed medication for adults covered by statutory health insurance [41]. Thus, over-indebted patients may be bound to weigh competing financial commitments and spending on necessities, such as (various) medications owing to co-payments. In line with these findings, this explorative study provides only a rough estimate of mental illness underlying antidepressant use among the over-indebted in Germany, and might underestimate the association between over-indebtedness and antidepressant use.

A key question that is unanswered yet is which causal mechanisms link over-indebtedness, mental illness and health care utilization. Legal and material consequences as well as stigmatization related to over-indebtedness may affect health both directly and indirectly via physiological processes and health-related behaviours. Over-indebtedness can reflect a source of chronic stress and ongoing worry which induce feelings of hopelessness and failure and lead to a decrease in mental health [6,11,14]. Since available resources may be largely allocated to debt repayment, the over-indebted may lack physical, mental and material capacities to maintain health and struggle to cover costs of health care and medications [6]. Moreover, due to decreased labour market participation and decreased workplace productivity, those suffering from a mental illness may be at greater risk of reduced income and job loss, respectively, that can in turn cause over-indebtedness [42].

A few limitations need to be acknowledged when interpreting the findings of this study. Due to the cross-sectional nature of the present study, the causal direction of the observed association remains unclear. It is both possible that mental illness underlying antidepressant use results from over-indebtedness and mental illness contributes to over-indebtedness by reducing the capability to manage debt [11,6,7]. Furthermore, a broadly accepted definition of over-indebtedness is not yet available [1]. Therefore, we defined OID participants that were seeking debt advisory agencies as over-indebted and assumed DEGS1 participants not to be over-indebted due to lack of data on indebtedness. However, it is likely that the DEGS1 population also includes individuals that are over-indebted according to this definition. This procedure might have yielded attenuated estimates of the association between over-indebtedness status and antidepressant use. Moreover, it cannot be excluded that the over-indebted who neither seek debt advice nor medical services are underrepresented in the OID survey. This, in turn, might overestimate antidepressant use in this population whereas CRN might have the opposite effect.

Given that antidepressants are mostly available only on prescription, the advantage of this measure is that it specifically takes underlying mental illnesses into account that have been diagnosed by a clinician. However, it is not possible to distinguish indications for antidepressant use or identify unmet need based on available data. Finally, the processes of collection of data on medication use differed between the two survey populations. In contrast to DEGS1,

OID relied solely on self-reporting which might have led to minor under-reporting. However, several studies suggest high validity of self-reported use of various medication types [43], such as antidepressants [44,45]. Despite limitations, this study provides first evidence of an association between over-indebtedness and antidepressant use and enhances the scientific understanding of socioeconomic inequalities in mental health.

## Conclusions

Depression is one of the leading causes of disability. In view of millions of over-indebted households across high-income countries, a growing body of evidence suggests higher morbidity among those affected. In support of previous research on socioeconomic inequalities in mental health, the present study is the first to show that over-indebted individuals are more likely to use antidepressants compared to the general population in Germany. Over-indebtedness thus seems to reflect an important public health issue. The findings emphasize the need to consider the potential risk of mental illness associated with over-indebtedness in health care, debt counselling, research and social policy. More specifically, a multidisciplinary approach is an essential prerequisite to facilitate effective and accessible treatment of widespread mental disorders that integrates both pharmacological and non-pharmacological intervention according to need. Further research is necessary to examine the causal mechanisms that may underlie the association between over-indebtedness and mental health in order to advance the understanding of health inequalities–beyond standard measures of socioeconomic status.

## Acknowledgments

Special acknowledgement is due to the staff at each debt advice agency in the Federal State of North Rhine-Westphalia, Germany, for their support in data collection.

## Author Contributions

**Conceptualization:** Judith Tillmann, Ulrike Zier, Klaus Weckbecker, Eva Münster.

**Data curation:** Jacqueline Warth, Niklas Beckmann, Marie-Therese Puth, Judith Tillmann, Johannes Porz, Eva Münster.

**Formal analysis:** Jacqueline Warth, Niklas Beckmann, Marie-Therese Puth.

**Funding acquisition:** Ulrike Zier, Klaus Weckbecker, Eva Münster.

**Methodology:** Jacqueline Warth, Niklas Beckmann, Marie-Therese Puth, Judith Tillmann, Ulrike Zier, Klaus Weckbecker, Eva Münster.

**Project administration:** Jacqueline Warth, Judith Tillmann, Ulrike Zier, Eva Münster.

**Supervision:** Klaus Weckbecker, Eva Münster.

**Validation:** Jacqueline Warth, Niklas Beckmann, Marie-Therese Puth, Johannes Porz.

**Writing – original draft:** Jacqueline Warth, Niklas Beckmann.

**Writing – review & editing:** Jacqueline Warth, Niklas Beckmann, Marie-Therese Puth, Judith Tillmann, Johannes Porz, Ulrike Zier, Klaus Weckbecker, Birgitta Weltermann, Eva Münster.

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
