## [Decision Letter · Decision Letter 0]

9 Apr 2020

PONE-D-19-32005

Association between Over-indebtedness and Antidepressant Use: A Cross-sectional Analysis

PLOS ONE

Dear Mrs. Warth,

Thank you for submitting your manuscript to PLOS ONE. After careful consideration, we feel that it has merit but does not fully meet PLOS ONE’s publication criteria as it currently stands. Therefore, we invite you to submit a revised version of the manuscript that addresses the points raised during the review process.

We would appreciate receiving your revised manuscript by May 24 2020 11:59PM. To enhance the reproducibility of your results, we recommend that if applicable you deposit your laboratory protocols in protocols.io, where a protocol can be assigned its own identifier (DOI) such that it can be cited independently in the future. For instructions see: http://journals.plos.org/plosone/s/submission-guidelines#loc-laboratory-protocols

We look forward to receiving your revised manuscript.

Kind regards,

Joel Msafiri Francis, MD, MS, PhD

Academic Editor

PLOS ONE

Journal Requirements:

Reviewers' comments:

Reviewer's Responses to Questions

**Comments to the Author**

1. Is the manuscript technically sound, and do the data support the conclusions?

Reviewer #1: Yes

Reviewer #2: Partly

2. Has the statistical analysis been performed appropriately and rigorously? 

Reviewer #1: Yes

Reviewer #2: No

3. Have the authors made all data underlying the findings in their manuscript fully available?

Reviewer #1: Yes

Reviewer #2: Yes

4. Is the manuscript presented in an intelligible fashion and written in standard English?

Reviewer #1: Yes

Reviewer #2: Yes

5. Review Comments to the Author

Reviewer #1: Reviewer comments

The article stablishes a correlation between over-indebtedness and antidepressant use using cross-sectional analysis.

Over-indebtedness is inferred by presence in the OID survey which comprises clients of debt advice agencies. (independent variable)

Antidepressant use is seen as a proxy for mental-health disorders like depression/anxiety but also sleeping disorders and chronic pain. (dependent variable)

Comments:

Antidepressant use can both be interpreted as presence of mental disorder and access to health-care.

When being in debt compromises access to health-care or medication, association between mental health need and actual use of medication can be compromised.

In lines 229 to 233 the authors give a brief explanation about how health-care and medication is provided in Germany. The text should make it clearer that being in debt doesn’t get in the way of access to health care and medication. Otherwise those who are worst off wouldn’t report antidepressant use and the inference would be compromised.

“In a cross-sectional survey among clients of debt advice agencies in Germany 65.2 percent of participants had not filled a prescription due to cost in the past year [39]. In Germany, co-payments of 5 to 10 euros are required for each prescribed medication for adults covered by statutory health insurance [40]. Thus, the actual need for antidepressants might be underestimated in the present study.”

It mentions that being in debt does compromise complience to treatment in most cases (65.2%). The author should explain how this doesn’t compromise the premise the reporting antidepressant use in the survey is linked to mental health suffering. Not only the actual need for antidepressant use can be underestimated but it could also go the opposite direction.

Databases

The authors merge two separate databases. One from the OID Survey which comprises a sample of clients of debt advice agencies and assumes that the whole of the sample is over-indebted. The survey took place between July and October 2017 in North-Rhine Westphalia (NRW). (n=699)

The other database comes from a nation wise survey in Germany (German Health Interview and Examination Survey for Adults (DEGS1)) with data colected betweew November 2008 and 2011 (n=7115).

What should be clarified is if data used from the DEGS1 comprises the whole Germany or of just the NRW part of the sample. Using a NRW sample from the DEGS1 and comparing it with the OID sample, would reduce statistical power but make the samples more comparable.

The authors should also clarify if they can be sure there’s no overlapping of the databases since they the data was collected independently, (i.e. – The same person being interviewed in bith surveys). And if there can be overlap how this does not interfere with the results.

Reviewer #2: This study examined the association between over-indebtedness and antidepressant use. Considering previous studies have reported the relationship between over-indebtedness and mental well-being, the authors should address and what the current study adds to what is already known in the Introduction

The study examined the harmful impact of over-indebtedness, but it is not clear why the authors used antidepressant use as the outcome measure. Antidepressant use is better measure for mental well-being? Antidepressant use is related to both presence of mental illness and access to mental health care. Over-indebtedness may affect negatively mental wellbeing but reduce access to medicine. The impact of over-indebtedness on mental well-being could be underestimated. Otherwise, did this study aim to examine access to medicine in people with mental illness due to over-indebtedness? Then, mental illness should be adjusted for in the statistical model. The analytical model depends on research question, however, which is not clear. Further, it the implications of the finding is not clear.

Over-indebtedness was defined with OID survey, information from clients of debt advice agencies. All people in over-indebtedness are included in the OID survey? I wonder people who seek more help are more likely to be clients of the agencies and to be prescribed with antidepressant. It should be described whether and how using OID survey to define over-indebtedness has advantages.

6. PLOS authors have the option to publish the peer review history of their article (what does this mean?). If published, this will include your full peer review and any attached files.

Reviewer #1: Yes: Márcio Souto de Castro Longo

Reviewer #2: No

---

## [Author Response · Author response to Decision Letter 0]

8 May 2020

Please kindly find the attached file "Response to reviewers".

---

## [Decision Letter · Decision Letter 1]

8 Jul 2020

Association between over-indebtedness and antidepressant use: A cross-sectional analysis

PONE-D-19-32005R1

Dear Dr. Warth,

We’re pleased to inform you that your manuscript has been judged scientifically suitable for publication and will be formally accepted for publication once it meets all outstanding technical requirements.

Kind regards,

Joel Msafiri Francis, MD, MS, PhD

Academic Editor

PLOS ONE

Additional Editor Comments (optional):

Reviewers' comments:

Reviewer's Responses to Questions

**Comments to the Author**

1. If the authors have adequately addressed your comments raised in a previous round of review and you feel that this manuscript is now acceptable for publication, you may indicate that here to bypass the “Comments to the Author” section, enter your conflict of interest statement in the “Confidential to Editor” section, and submit your "Accept" recommendation.

Reviewer #1: All comments have been addressed

2. Is the manuscript technically sound, and do the data support the conclusions?

Reviewer #1: Yes

3. Has the statistical analysis been performed appropriately and rigorously? 

Reviewer #1: Yes

4. Have the authors made all data underlying the findings in their manuscript fully available?

Reviewer #1: Yes

5. Is the manuscript presented in an intelligible fashion and written in standard English?

Reviewer #1: Yes

6. Review Comments to the Author

Reviewer #1: (No Response)

7. PLOS authors have the option to publish the peer review history of their article (what does this mean?). If published, this will include your full peer review and any attached files.

Reviewer #1: **Yes: **Márcio Souto de Castro Longo

---

## [Editor Report · Acceptance letter]

10 Jul 2020

PONE-D-19-32005R1 

Association between over-indebtedness and antidepressant use: A cross-sectional analysis 

Dear Dr. Warth:

I'm pleased to inform you that your manuscript has been deemed suitable for publication in PLOS ONE. Congratulations! Your manuscript is now with our production department. 

Kind regards, 

on behalf of

Dr. Joel Msafiri Francis 

Academic Editor

PLOS ONE